# Genome-Wide Identification and Characterization of Long Non-Coding RNAs in Peanut

**DOI:** 10.3390/genes10070536

**Published:** 2019-07-15

**Authors:** Xiaobo Zhao, Liming Gan, Caixia Yan, Chunjuan Li, Quanxi Sun, Juan Wang, Cuiling Yuan, Hao Zhang, Shihua Shan, Jian Ning Liu

**Affiliations:** 1Shandong Peanut Research Institute, Qingdao, Shandong Province 266000, China; 2KeGene Science & Technology Co. Ltd., Tai’an, Shandong Province 271000, China

**Keywords:** peanut, lncRNA, RNA sequencing, WGCNA

## Abstract

Long non-coding RNAs (lncRNAs) are involved in various regulatory processes although they do not encode protein. Presently, there is little information regarding the identification of lncRNAs in peanut (*Arachis hypogaea* Linn.). In this study, 50,873 lncRNAs of peanut were identified from large-scale published RNA sequencing data that belonged to 124 samples involving 15 different tissues. The average lengths of lncRNA and mRNA were 4335 bp and 954 bp, respectively. Compared to the mRNAs, the lncRNAs were shorter, with fewer exons and lower expression levels. The 4713 co-expression lncRNAs (expressed in all samples) were used to construct co-expression networks by using the weighted correlation network analysis (WGCNA). LncRNAs correlating with the growth and development of different peanut tissues were obtained, and target genes for 386 hub lncRNAs of all lncRNAs co-expressions were predicted. Taken together, these findings can provide a comprehensive identification of lncRNAs in peanut.

## 1. Introduction

Non-coding RNA is a major transcript produced by eukaryotic genome, which is different from mRNA [1]. With respect to their expression characteristics, non-coding RNAs can be broadly classified into housekeeping non-coding RNAs and regulatory non-coding RNAs. The housekeeping non-coding RNAs, such as rRNAs, small nuclear RNAs (snRNAs), tRNAs, and small nucleolar RNAs (snoRNAs) are essential for maintaining the basic functions of cells. The regulatory non-coding RNAs can be particularly expressed in specific tissues during different developmental stages of plants or after biotic and abiotic stresses [2]. They can be further classified into small (<200 base pairs (bp) and long non-coding RNAs (lncRNAs; >200 bp) based on their lengths. The functions and mechanisms of small non-coding RNAs have been completely elucidated over past decades; however, lncRNAs have not been extensively studied [3,4,5]. Recent studies have reported that lncRNA plays an important role in many biological processes, including cell growth and proliferation, intracellular processes, developmental processes, disease resistance pathways, and transcriptional regulation [6,7,8,9,10]. For example, a class of lncRNAs plays an unanticipated role in the activation of critical regulators associated with development and differentiation in human cells [6].

Interestingly, lncRNAs are related to diverse mechanisms of biological processes, although they lack obvious open reading frames (ORFs), which may be related with the fact that the expression of target genes is regulated by the lncRNAs [8,11]. Target genes of lncRNA are predicted mainly based on their roles—*cis* or *trans*. In *cis* role, the function of lncRNA is related to adjacent protein-coding gene. The lncRNA located upstream and downstream of the protein-coding gene may regulate the gene expression at the transcriptional or post-transcriptional levels. In *trans* role, The function of lncRNA is related to co-expressed protein-coding gene. When lncRNA is positively or negatively correlated with the expression levels of some genes, the target gene can be predicted by correlation analysis or co-expression analysis of lncRNAs and protein-coding genes between samples. Prediction of target gene in this *trans* role can be achieved by performing the weighted correlation network analysis (WGCNA) [12].

In yeast and other higher eukaryotes, a large number of lncRNAs have been discovered [13,14], such as more than 50,000 lncRNAs in the human genome discovered by genome-wide analyses [15,16]. Presently, a number of lncRNAs have been identified by high-throughput sequencing in many plants. In general, lncRNAs demonstrate low expression, lack the conservation between species, and often exhibit cell-specific or tissue-specific expression patterns [11,15]. In *Arabidopsis*, more than 6000 intergenic lncRNAs were reported [11] and different expression levels of lncRNAs were observed under various stress stimuli [17]. In *Oryza sativa* L., lncRNAs associated with sexual reproduction were identified [18]. In *Zea mays* Linn., drought-responsive lncRNA was identified [19], including a comprehensive set of maize lncRNAs [20]. In *Gossypium* spp., several functional lncRNA candidates involved in cotton fiber initiation and elongation were identified and the first comprehensive identification of lncRNAs was reported [21]. In *Lycopersicon esculentum* Mill., lncRNAs that responded to TYLCV infection were identified and competing endogenous target mimics (eTMs) for tomato microRNAs involved in the TYLCV infection were regulated by several lncRNAs [22]. In addition, novel insights into the evolution of lncRNAs were provided in *Solanum lycopersicum* L., and lncRNAs played an important role in fruit ripening [23]. These genome-wide studies mainly focused on the identification of unknown lncRNAs and on providing their expression patterns.

In this study, we used large-scale published RNA sequencing data, including 124 peanut samples from 15 different tissues to identify and characterize genome-wide lncRNAs in peanut. A total of 50,873 lncRNAs was obtained from all unknown transcripts. The co-expression modules were generated by performing WGCNA, and 386 hub lncRNAs in different modules were obtained. In addition, the potential functions of hub lncRNAs were predicted by using *cis* methods and 134 known target genes and 252 unknown target genes were obtained. These results can provide a rich resource of genome-wide lncRNAs in peanut.

## 2. Materials and Methods

### 2.1. Publicly Available Data Sets Used in This Study

RNA data sets of *A. hypogaea* were obtained from the Sequence Read Archive of the National Center for Biotechnology Information (NCBI) collection. These RNA data sets were related to five major organizations from 124 RNA-seq experiments, including flower (flowers, gynophore tip, gynophore stalk, pistils, and stamens), stem (nodules and reproductive and vegetative shoots), leaf, fruit (pericarp, pod, pod wall, post-harvest seed, and seed), and root (Appendix A).

### 2.2. LncRNA Identification and Target Gene Prediction

The quality of all RNA sequences was examined by using FASTQC (v. 0.10.1) (www.bioinformatics.bbsrc.ac.uk/projects/fastqc). The adaptors and low-quality bases were removed from the sequence using Trimmomatic (v. 0.39) [24]. Reads with a phred quality (Q > 30) were used for further alignment and assembly. All clean reads from each experiment were aligned to the diploid peanut reference genome released in 2012 [25] using spliced read aligner HISAT (v. 2.1.0) [26]. The transcripts of each experiment were assembled separately using StringTie (v. 1.3.4d) [26]. To reduce transcriptional aberration, the assembled transcript isoforms were detected in at least two or more samples for subsequent analysis. Gffcompare (v. 0.11.2) was used to compare the assembled transcript isoforms with peanut genome annotation [26]. A total of 308,565 transcripts, including 152,156 known transcripts and 156,409 unknown transcripts were generated by the above processes.

Later, FEELnc (v. 0.1.1) was used to identify lncRNAs from unknown transcripts of transcriptome assemblies. In the initial step, FEELncfilter was used to filter out short transcripts (default 200 nt) and assess single-exon transcripts [27]. In the next step, FEELnccodpot predictors were used, aiming to compute a coding potential score considering that the assembled sequences were provided following the transcriptome reconstruction. Finally, RNAs meeting the properties of length ≥ 200 bp, ORF cover < 50%, and potential coding scores < 0.5 are defined as lncRNAs. The functions of hub lncRNAs were explored by predicting the target genes of hub lncRNAs in *cis* role using FEELncclassifier. Genes located at 100 kb upstream or downstream of lncRNAs were defined as target genes of lncRNAs [20,28].

### 2.3. Abundance and Length Distributions

The abundance of all assembled transcripts was expressed as fragments per kilobase of exon model per million mapped base pairs (FPKM). To get average FPKM, the abundance in all lncRNAs and mRNAs of all non-zero samples was averaged. Then, the average FPKM was transformed by log10, and the log2 FPKM distribution was generated by performing the R function density based on kernel density. The R (plot(), lines() and legend()) was used to plot distributions of these kernel density. The length frequency distribution is generated by performing R(hist()).

### 2.4. Co-Expression Modules

The co-expression modules were generated using WGCNA (v. 1.67) [29]. To acquire all available information, lncRNAs that were not expressed in at least one tissue were not considered for this analysis. In each network, the parameters were defined. In all networks and two correlation methods, number 18 was the satisfactory soft power threshold. Adjacency matrices were constructed using the adjacency() function according to soft power threshold. According to adjacency matrix, a topological overlap (TO) matrix was generated based on TOM similarity algorithm and then lncRNAs were hierarchically clustered based on the algorithm. Dynamic tree-cutting algorithm was adopted to obtain hierarchical clustering dendrogram. To achieve a stable number of clusters, modules were defined after decomposing or combining branches. The summary profile was calculated via PCA in each module. Further, the modules with higher TO value (average TO for all lncRNAs in a given module) were retained after comparing the TO values of modules that included randomly selected genes.

### 2.5. Identification of Tissues-Specific Modules and Construct Co-Expression Networks

To determine the association of modules with different tissues, we determined the modules with (*r* > 0.75) as tissues-specific modules. A positive correlation indicated that lncRNAs in a module have higher/preferential expression in a particular tissue relative to all other tissues. The co-expression networks of lncRNAs and hub lncRNAs in high correlation modules were generated using CentiScaPe (v. 2.2) in cytoscape software (v. 3.7.1) [30].

## 3. Results

### 3.1. Genome-Wide Identification of LncRNAs in Peanut

To identify a relatively large set of lncRNAs of peanut, the reads were mapped to the peanut reference genome and assembled into 308,565 transcripts (132,039 genes) (Figure 1a). In total, there were 152,156 known transcripts (98,072 known genes) and 156,409 unknown transcripts (33,968 unknown genes). A total of 50,873 lncRNAs (16.47%) (length ≥ 200 bp, ORF cover < 50%, potential coding scores < 0.5) were obtained from unknown transcripts by performing FEELnc. In addition, 101,033 mRNAs (32.74%) were generated from known transcripts (Figure 1b; Appendix A). The lengths of 85% mRNAs ranged from 1000 to 10,000 bp; however, the majority of lncRNAs (67%) were relatively short ranging from 200 to 1000 bp (Figure 1c,d), suggesting that the sequence components of the mRNA exhibit differently when compared to the lncRNAs.

### 3.2. Characterization of Peanut LncRNAs

The distributions of lncRNAs and mRNAs on peanut chromosome were measured by using Circlize (v. 0.2.5) [31]. Lesser densities of lncRNAs were observed than chromosome “arms” in the pericentromeric regions of most peanut chromosomes, while peanut lncRNAs were found to be more evenly distributed across all chromosomes (Figure 2a). The characterization analysis of lncRNAs was performed according to the locations relative to the nearest protein-coding genes. The number of lncRNAs located in intergenic regions was 33,521 (65.89%) and only 34.11% of lncRNAs overlapped with gene sequences (Figure 2b). The number of intergenic lncRNAs located within upstream and downstream of genes were found to be 6781 (20.2%) and 7519 (22.4%), respectively (Figure 2b). The remaining 57.4% of intergenic lncRNAs were at least 5 kb away from the nearest gene. Approximately 52.92% of lncRNAs were composed of a single exon, and the remaining had multiple exons (Figure 2c). However, the number of exons in mRNAs was significantly higher than that of the lncRNAs (Appendix A).

### 3.3. Expression Level of LncRNA in Different Tissues

The tissue-specificity of lncRNA expression was examined by using the transcriptome data from 124 different samples that represented 5 major tissue types. Different organizations involve different numbers of samples, such as root (23 samples), fruit (54 samples), leaf (10 samples), stem (9), and flower (16). In addition, 12 samples were found to contain the mixed sequencing of pod, leaves, and roots, and were defined as co-expressed transcripts of fruit, leaf, and root (Appendix A). Numerous lncRNAs (20,106) were detected in all tissues, with 12.1% of lncRNAs detected in only one tissue, including 3885 lncRNAs in root, 102 lncRNAs in flower, 2044 in fruit, 46 lncRNAs in leaf, and 58 lncRNAs in stem (Figure 3a). Analysis of the expression levels for all five tissues provided evidence that the lncRNAs of stem tend to exhibit lower expression than other tissues (Figure 3b). The average expression level between lncRNAs and mRNA was also analyzed, and it was found that the expression level of lncRNAs was lower than that of mRNA (Appendix A). Analysis of tissue-specific expression indicated that the expression of these sequences (lncRNAs) was biologically controlled instead of being attributed to “transcriptional noise.”

### 3.4. Identification of LncRNAs Co-Expression Modules

In addition, we examined the patterns of lncRNAs across all peanut tissues (leaf, root, pericarp, pod, pod wall, post-harvest seed, seed, nodules, reproductive shoot, vegetative shoot, flowers, gynophore tip, gynophore stalk, pistils, and stamens) involved in transcriptome data. WGCNA was used to group the genes into modules (clusters) according to similarly expressed lncRNAs. To avoid the inclusion of false edges in the networks, lncRNAs that were not expressed in at least one tissue were excluded in this analysis. Finally, 4713 lncRNAs were generated for further analysis, with the identification of 11 modules (61–1990 lncRNA) (Figure 4a). In addition, Pearson correlation coefficient analysis was performed to connect each of the co-expression modules with peanut tissues. It was observed that various tissues was correlated with only one co-expression module, such as the turquoise module specifically correlated (*r* = 0.99) with leaf, green module with seed (*r* = 0.98), and pink module with root (*r* = 0.94) (Figure 4b). Besides, the turquoise module contained the largest number of lncRNAs (Figure 4c). Later, nine highly correlated modules (*r* > 0.75) were used to construct co-expression network that included all modules, such as turquoise (leaf, 0.99), pink (root, 0.94), green-yellow (flowers, 0.78), yellow (stamens, 0.85), red (nodules, 0.92), black (pericarp, 0.79), brown (pod wall, 0.94), blue (post-harvest seed, 0.94), and green (seed, 0.98). In these 9 modules, the correlation between the module associations and lncRNAs significance was demonstrated for tissues and expression profiles (Appendix A).

### 3.5. Potential Functional Roles for Peanut LncRNAs

Hub lncRNAs are the most important elements of the network in each module that are essential in the regulation of networks. Thus, CentiScaPe 2.2 was used to identify and obtain the hub lncRNAs in these 9 modules. Hub lncRNAs were generated according to node degree centrality and the number of hub genes ranging from 26 to 60 in different modules (Appendix A). To investigate the potential functional roles or biological processes that these hub lncRNAs might be involved in, their target genes were examined. Genes located at 100 kb upstream or downstream of lncRNAs were defined as target genes of lncRNAs [20,28]. Results indicated that a total of 386 target genes were obtained, including 134 known target genes and 252 unknown target genes (Appendix A). In turquoise module, 29 hub lncRNAs may be related to the growth and development of peanut leaves. The 11 known target genes for 29 hub lncRNAs were found to be involved in multiple functions, such as calmodulin-like protein 7, zinc finger MYM-type protein 1, and F-box protein (Figure 5a). Further, 56 hub lncRNAs were generated in red module that were related to nodules (stem) and their 12 known target genes were involved in many functions, such as UDP-glycosyltransferase and splicing factor (Figure 5b). In addition, the target genes encoding early nodulin- and auxin-induced proteins were found in the root-related pink module (Figure 5c). Moreover, the target genes of other modules exhibited high correlation with different peanut tissues and were found to be involved in multiple functions (Figure 6). These results suggested that lncRNAs may regulate the growth and development of the corresponding tissues by acting on their target genes.

## 4. Discussion

Transcriptome consists of protein-coding genes and some specific non-coding RNAs. However, with the advancement in bio-technology, it has been observed that the expression of eukaryotic genome is far more complex than previous findings [32,33]. The non-protein coding genes, such as long non-protein coding genes and short non-protein coding genes (small interfering RNAs and miRNAs), are essential for the complexity of eukaryotic genome. The function of short non-coding RNAs in transcriptional and post-transcriptional regulation of target gene expression has been completely elucidated [34]. However, little information is available about the lncRNAs with respect to their functions and characteristics in various plant species. Previous studies have confirmed that lncRNAs played an important role in growth and development [35].

In this study, 124 published RNA-seq data sets from different tissues of different cultivars of peanuts from NCBI were obtained and 308,565 transcripts were generated, including 152,156 known transcripts (98,072 known genes) and 156,409 unknown transcripts (33,968 unknown genes). Later, there were 50,873 (16.47%) lncRNAs from unknown transcripts obtained by performing FEELnc and 101,033 (32.74%) mRNA from known transcripts were generated. The sequence lengths of lncRNAs are shorter, exon numbers are fewer, and lower expression levels compared to those of mRNAs, which was consistent with previous studies on other organisms [15,36].

All the RNA-seq data involved in five major organizations were divided into 15 tissues. To obtain the co-expression modules related to these 15 tissues, the WGCNA was performed on 4713 co-expression lncRNAs (expressed in all samples). A total of 11 co-expression modules were generated and 9 of the 11 modules are highly correlated (*r* > 0.75) with different modules, respectively. A total of 386 hub lncRNAs were generated from these 9 modules. Subsequently, the potential functions of the hub lncRNAs were predicted by using *cis* methods and 134 known target genes and 252 unknown target genes were obtained.

In turquoise module (Figure 5a) associated with leaf, the target genes (LOC112803798 and LOC112797235) of hub lncRNAs (MSTRG.21437.1 and MSTRG.978.1) are involved in F-box protein and HMG1/2-like protein, respectively, which affect leaf size in *Arabidopsis* [37] and play an important role in the development in *Triticum aestivum* L. [38]. Target gene (LOC112701746) of hub lncRNA (MSTRG.3167.1) encodes calmodulin-like protein 7. It was reported that the interaction of mildew resistance locus O (*CsMLO1*) and calmodulin (*CsCaM3*) genes may be associated with plant immunity by acting as a cell death regulator [39], suggesting that hub lncRNA (MSTRG.3167.1) may participate in plant immunity. In pink module (Figure 5c) associated with root, the target gene (LOC112696267) of hub lncRNA (MSTRG.24254.1) encodes auxin-induced protein 22E, and the directional transport of auxin plays an important role in the formation of adventitious roots [40] and auxin promotes root growth [41], suggesting that lncRNA MSTRG.24254.1 may promote the growth of peanut roots. U-box domain-containing protein 3 (LOC112727061), as a target gene of hub lncRNA MSTRG.47968.1, can combat drought stress and pathogen infection in barley [42] and regulate growth, development, and stress responses in plants [43]. In modules related to fruit (black module (Figure 6a) associated with pericarp, brown module (Figure 6c) associated with pod wall, blue module (Figure 6b) associated with post-harvest seed, and green module (Figure 6d) associated with seed), the target gene (LOC112787272 in black module (Figure 6a) and LOC112736609 in brown module (Figure 6c)) of hub lncRNAs encodes histone H2B, which is associated with activation and regulation of methyltransferases [44]. Target gene LOC112730895 of hub lncRNA MSTRG.41333.1 encodes ABC transporter C family member 15 (Figure 6a). It was reported that ABA homeostasis contributes to drought tolerance through a balance between the production, catabolism and transport in peanut leaves [45]. Target gene LOC112726802 of hub lncRNA MSTRG.47603.1 encodes BON1-associated protein 2, which is a plasma membrane-localized protein, plays a negative role in regulating the expression of immune receptor genes, but has a positive regulation in stomatal closure [46]. In addition, auxin-responsive protein IAA is also related to target gene LOC112695655 of hub lncRNA MSTRG.23480.1 in green module (Figure 6d). In red module (Figure 5b) associated with stem, the target genes of hub lncRNAs involved multiple functional target genes, such as UDP-glycosyltransferase and transcription factor GTE9. UDP-glycosyltransferase (UFGT2) contributes to improving the tolerance to abiotic stresses by modifying flavonols in maize. ABA and sugar responses are mediated by the interaction of Global Transcription Factor Group E proteins GTE9 and GTE11 and BT2 in *Arabidopsis* [47]. Interestingly, the target gene LOC112696596 encoding protein shoot gravitropism 6 was only found in red module (Figure 5b), suggesting that this protein may play an important role in peanut nodule. In modules related to flower (green-yellow module (Figure 6e) associated with flowers and yellow module (Figure 6f) associated with stamens), the target genes of hub lncRNAs were involved in multiple functional types, such as pathogenesis-related protein PRB1, WRKY transcription factor 33, and auxin-induced protein AUX22. It was reported that an organ-specific expression pattern is established by AtPRB1, which responds to ethylene and methyl jasmonate in *Arabidopsis* [48]. Ethylene response factor (ERF) transcription factor, VaERF092, from Amur grape binds the promoter GCC-box of VaWRKY33, leading to enhanced cold stress tolerance in *Arabidopsis* [49]. Together, these target genes of all hub lncRNAs were related to a number of functions, such as growth and development and response to abiotic and biotic stress, suggesting that lncRNAs play multiple functions in peanut. These findings may provide a valuable theoretical basis for cultivating high-yield and high-resistance peanut cultivar.

## 5. Conclusions

In this study, we obtained 50,873 lncRNAs from 124 published RNA-seq data sets involving 15 peanut tissues. Compared with the mRNA, the lncRNAs sequences are shorter, with lesser number of exons and lower expression levels. The co-expression modules for 4713 co-expressive lncRNAs (expressed in all samples) were generated by performing the WGCNA to investigate the relationship between lncRNAs and different tissues. The co-expression network for each module with high correlation with the tissues was constructed and target gene prediction was executed for hub lncRNAs. This may provide a mechanism of the mRNA expression regulation by the lncRNAs in different tissues of peanut.

## Figures and Tables

**Figure 1 genes-10-00536-f001:**
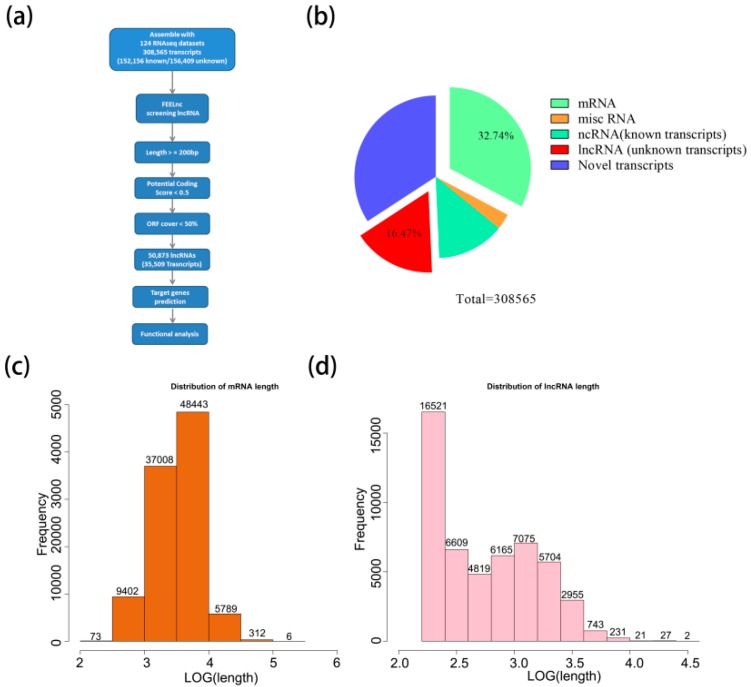
Identification and characterization of novel long non-coding RNAs (lncRNAs) in peanut. (**a**) Flow diagram of the bioinformatics pipeline for the identification of lncRNAs in peanut. The potential coding scores were calculated by FEELnc. (**b**) The proportion of lncRNA in all transcripts. (**c**,**d**) Length distribution of mRNA and lncRNA.

**Figure 2 genes-10-00536-f002:**
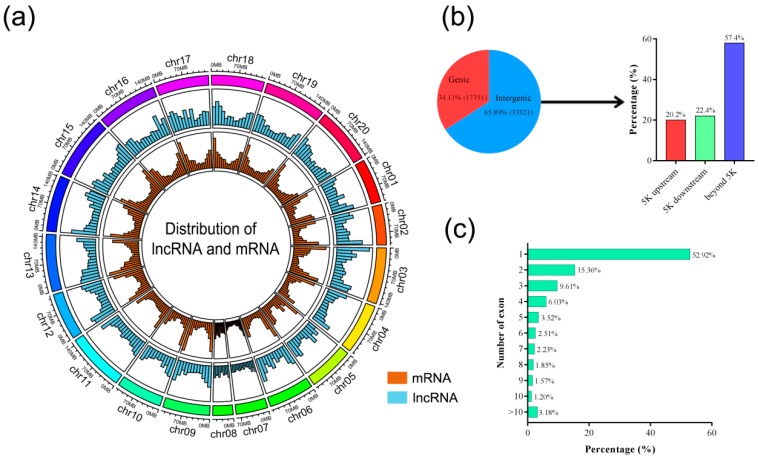
Characteristics of peanut lncRNAs. (**a**) Distribution of lncRNAs and mRNA along each chromosome. The abundance of lncRNA and mRNA in physical bins of 10 Mb for each chromosome (generated using Circlize). (**b**) Proportion of lncRNAs that are located within 5 kb (upstream or downstream) or further than 5 kb from the nearest protein-coding genes. (**c**) Numbers of exons in lncRNA.

**Figure 3 genes-10-00536-f003:**
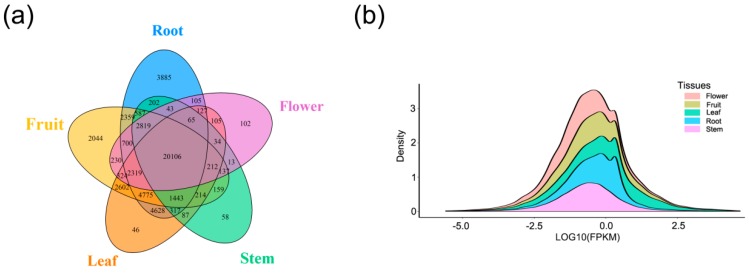
Tissue-specific expression and expression levels of lncRNAs. (**a**) The Venn diagram shows the overlap of lncRNAs between these five tissues. (**b**) Density plot for average expressions of lncRNAs in five tissues.

**Figure 4 genes-10-00536-f004:**
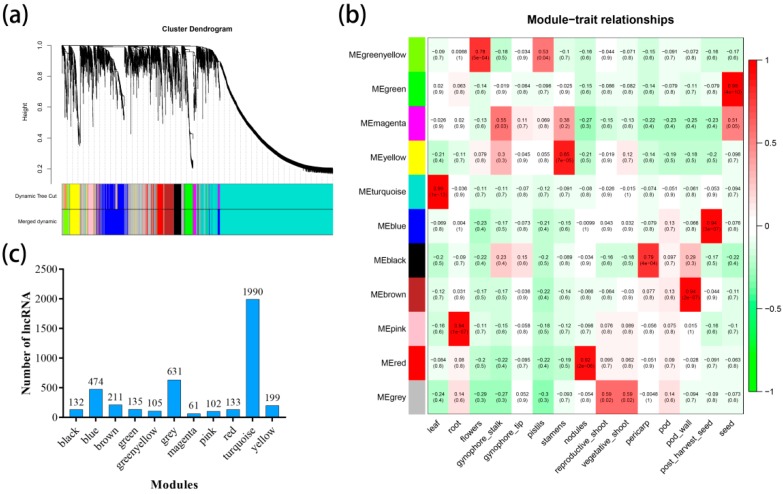
Weighted gene co-expression network analysis (WGCNA) of lncRNAs in all samples. (**a**) Hierarchical cluster tree showing 11 modules of co-expressed lncRNA. Each of the 4713 lncRNAs is represented by a leaf in the tree while each of the 11 modules by a major tree branch. The lower panel shows modules in designated colors. (**b**) Heatmaps showing correlation of module eigengenes with different tissues of peanut. Pearson correlation coefficient of each module with different tissues are given and colored according to the score. (**c**) Number of lncRNA in each module.

**Figure 5 genes-10-00536-f005:**
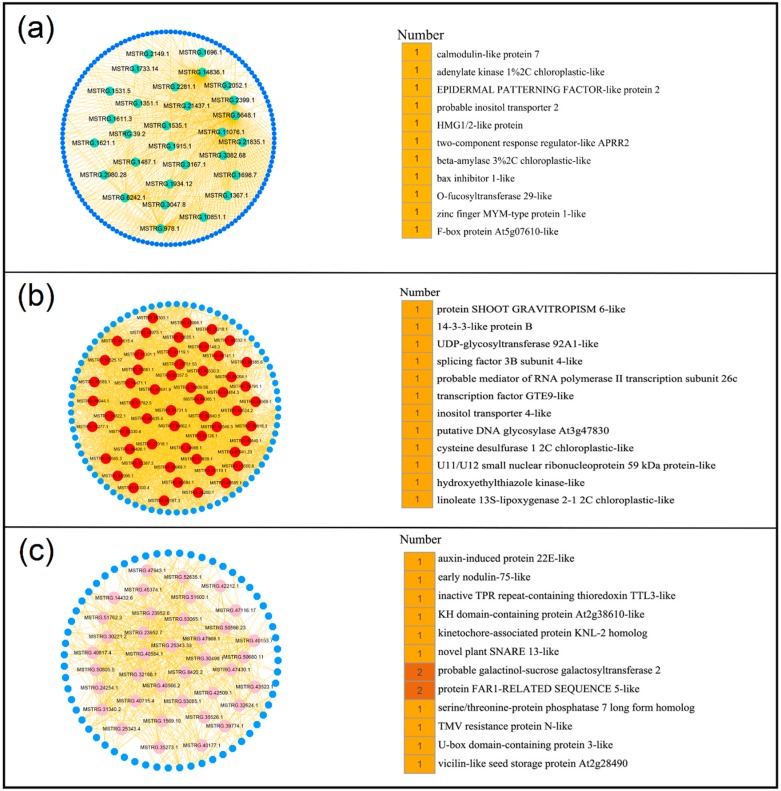
Constructing co-expression network associated with turquoise, pink, and red modules. (**a**) Turquoise module. (**b**) Red module. (**c**) Pink module. (**Left)** The network of hub lncRNA is shown in circular. (**Right**) Functional annotation of the target genes of the hub lncRNAs.

**Figure 6 genes-10-00536-f006:**
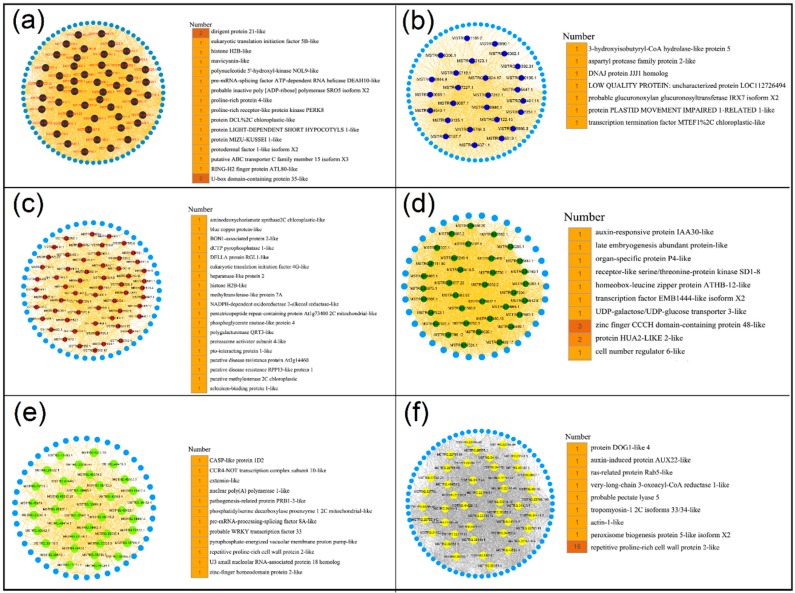
Constructing co-expression network associated with black, blue, brown, green, green-yellow, and yellow modules. (**a**) Black module. (**b**) Blue module. (**c**) Brown module. (**d**) Green module. (**e**) Green-yellow module. (**f**) Yellow module. (**Left)** The network of hub lncRNA is shown in circular. (**Right)** Functional annotation of the target genes of the hub lncRNAs.

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
