# Peer review of "Genome-Wide Identification and Characterization of Long Non-Coding RNAs in Peanut"

_genes, 2019, doi:10.3390/genes10070536_

Round 1

Reviewer 1 Report

The manuscript has important informations for peanut research community.   I have some suggestions to improve the text:

1)      You should give more details about the functional annotation of the target genes of the hub lncRNAs. For example, why we see protein shoot gravitropism 6-like in nodule module and we did not observe this in vegetative shoot or root module?

2)      You could insert a new table with the functions of each functional annotation of the known target genes of the hub lncRNAs and the relation among them into each module.

3)      What will be the next steps to use these informations in peanut biotechnology or plant breeding research?

 I suggest the publication of this research  with minor changes.

Reviewer 2 Report

·         Line # 12: In this study, 50,873 lncRNAs of peanut were generated.

Change this sentence to: In this study, 50,873 lncRNAs of peanut were identified/predicted.

·         The abstract is to be made clear by improving the language and providing some technical details. For example, the average length of the lncRNA can be mentioned.

·         Line # 42: Do you consider nat-siRNAs as lncRNA? If not, why to mention it?

·         In the introduction, briefly mention the functions of the lncRNAs instead of just citing the references.

·         Line # 59: How many genotypes were used in this study? On an average how many samples corresponded to each genotype?

·         Line # 112: What was the criteria used to identify the lncRNAs?

·         How many lncRNAs were commonly found in all the genotypes and samples? How many were unique to a genotype or sample?

·         Which DNA dependent RNA polymerase does synthesize lncRNAs?

·         Line # 190: How do you define the target gene for a lncRNA? Is it the gene which codes for lncRNA or the gene which is acted upon by lncRNA? Therefore, it will be nice if you define the functions of the lncRNAs in the introduction.

·         Try to draw the general inferences from the results of various experiments/analyses apart from making the specific ones.

·         What distinct properties of the lncRNAs did you use to identify them in this study?

·         Line # 62-69: Mention clearly which genome (tetraploid or diploid) you used as the reference for mapping the reads. Also cite the reference.
